# Effect of Physical Exercise on Executive Functions Using the Emotional Stroop Task in Perimenopausal Women: A Pilot Study

**DOI:** 10.3390/bs14040338

**Published:** 2024-04-18

**Authors:** Li-Yu Wu, Hsiu-Chin Hsu, Lee-Fen Ni, Yu-Jia Yan, Ren-Jen Hwang

**Affiliations:** 1Department of Nursing, Chang Gung University of Science and Technology, Taoyuan 333424, Taiwan; lywu@mail.cgust.edu.tw (L.-Y.W.); lfni@mail.cgust.edu.tw (L.-F.N.); goshassault@gmail.com (Y.-J.Y.); 2Graduate Institute of Gerontology and Health Care Management, Chang Gung University of Science and Technology, Taoyuan 333424, Taiwan; hchsu@mail.cgust.edu.tw; 3Department of Internal Medicine, Chang Gung Memorial Hospital, Taoyuan 333423, Taiwan; 4Department of Nursing, Linkou Chang Gung Memorial Hospital, Taoyuan 333423, Taiwan; 5Clinical Competency Center, Chang Gung University of Science and Technology, Taoyuan 333424, Taiwan; 6Intellectual Property Office, MOEA, Taipei City 100210, Taiwan

**Keywords:** exercise, perimenopausal women, women’s health, emotional Stroop task (EST), executive function

## Abstract

Exercise has beneficial effects on emotional cognitive control for the majority of the population. However, the impact of exercise on cognitive processes in perimenopausal women remains unclear. Therefore, this study investigated the effect of aerobic exercise on the cognitive processes of perimenopausal women using an emotional Stroop task (EST). Method: A quasi-experimental pilot study was conducted involving 14 perimenopausal women (Peri-MG) and 13 healthy young women (YG) who completed an EST before and after an aerobic cycling exercise. Mixed-effects models for repeated measures were used to analyze reaction times (RTs) and error rates (ERs) during emotional word processing (positive, negative, and neutral) for both groups. Results: Compared with the YG, the Peri-MG showed significantly shortened RTs for positive and negative emotions (*p* < 0.05) post-exercise, but not for neutral words. In addition, the Peri-MG exhibited significantly increased ERs for negative words at baseline compared with the YG (*p* < 0.05), but this difference was not observed during the post-exercise test. Conclusion: The findings suggest that aerobic exercise can enhance executive control performance in perimenopausal women. The Peri-MG exhibited marked behavioral plasticity in the form of reduced bias to salient cues that were significantly more sensitive to alterations due to exercise. This new evidence enhances the understanding of emotional vulnerability and beneficial susceptibility to exercise in perimenopausal women.

## 1. Introduction

### 1.1. Cognitive Function in Perimenopausal Women

The terms “menopausal transition” or “perimenopause” are used to describe the period in which women experience changes in reproductive hormone levels and menstrual cycle patterns. The perimenopausal period is a time of increased vulnerability to cognitive decline and elevated risk of depressive symptoms and mood disturbances [1,2,3,4]. Menopausal transition usually starts between 45 and 55 years of age in 95% of women, leading to the postulation that endogenous estrogen fluctuations are associated with physical and emotional changes and the exacerbation of psychiatric symptoms [5,6,7]. Several neurocognitive functions are modulated during the menstrual cycle, pregnancy, and throughout a woman’s entire lifespan. Reproductive hormones are involved in the function of the hippocampus and prefrontal cortex, as well as cognitive performance, including verbal, memory, and executive functions [7,8,9,10]. The general pattern of age-related preservation and decline indicates that prefrontal white matter is the most susceptible to the influence of age [2,3,8]. Age-related cognitive decline has also been reported during the menopausal period, usually in those aged 60–65 years; however, its extent remains undetermined in middle-aged perimenopausal women.

### 1.2. Exercise and Cognition

The effects of aerobic exercise on mental well-being and physical fitness have been extensively demonstrated [9,10]. Several studies have revealed the relationship between fitness and cognition in different populations [11]. Both acute and chronic exercises can significantly improve mood, behavior, and cognitive abilities [12], with positive effects on emotional cognitive control and function [13]. Evidence suggests that exercise is beneficial for activities of daily living because it influences reaction times (RTs). The effects of acute exercise on neural substrates coincide with improved cognitive performance [13]. It is well established that frontal neural circuits play a crucial role in mediating exercise-related changes in cognitive function and emotional processing [14,15]. The region-specific degenerative changes in prefrontal function that occur with increasing age have also been documented [16,17]. Given the rapid growth of aging populations in some countries, several studies have focused on age-related cognitive flexibility caused by exercise [18], mainly including the benefit of aerobic exercise on cognitive ability in individuals above 60 years in age [19,20]. However, there is a paucity of similar data on middle-aged perimenopausal women.

### 1.3. Emotional Stroop Tasks

The Stroop color-naming paradigm is a commonly used tool for examining cognitive control [21]. Similarly, an emotional Stroop task (EST) investigates the effects of emotions on cognitive control. An individual is given words that induce negative emotions (e.g., “death”) printed in one color (e.g., red), mixed with more neutral words, such as “desk”, printed in the same color, and/or positive words printed in a different color. The subject must name the color as rapidly and accurately as possible while simultaneously ignoring the meaning of the word. In general, processing the emotional content of words imposes a cognitive load that leads to delayed color naming [21]. Participants are usually slower when naming the color of emotionally laden words than those of neutral words [22]. This delay, reflected as longer RTs, is called the emotional Stroop effect (ESE). The ESE demonstrates a person’s sensitivity to the emotional valence of stimuli. One common explanation of the ESE is that the affective nature of emotionally laden words interferes with color-naming by capturing attentional resources [22]. The EST also measures how well an individual can maintain goal-oriented processing. The ability to sustain mental processes and select appropriate tasks inhibits the relatively automatic word-reading response as a specific executive function [23,24,25].

The emotional interloping and cognitive decline observed during perimenopause are closely associated [26]. Meanwhile, the effects of aerobic exercise on cognition and attention reduce the ESE [27,28,29]. EST can be used to assess emotions and has evolved as a method to measure anxiety in both patient and nonpatient populations. Significantly, anxiety- or depression-prone participants exhibit attentional bias and alterations in their cognitive function [30,31].

Recently, exercise has garnered attention as a means of improving neurocognitive function and facilitating healthy aging. Further research is needed to understand the mechanism by which exercise affects middle age-related changes in cognitive control. However, a simple EST can be used to explore the aerobic effects of exercise on executive function in Peri-MG. This study aims to compare the effectiveness of aerobic exercise in perimenopausal women with that in young women using a modified EST. We hypothesize that perimenopausal women may demonstrate a different cognitive processing performance on the EST (RTs and error rates, or ERs) compared with young women.

## 2. Materials and Methods

### 2.1. Study Design and Participants

This study adopts a quasi-experimental design. Participants were recruited through purposive sampling, with 14 perimenopausal women (Peri-MG; aged 46–54 years) recruited who were eligible following the latest Stages of Reproductive Ageing Workshop (STRAW) criteria, which are based on self-reported bleeding patterns [32]. Thirteen healthy young women (YG; aged 18 to 22 years) were recruited among local university students. All participants completed two neuropsychological assessments: (1) a 20-item state anxiety questionnaire (AQ) [33], in which each question was given a weighted score of 1 to 4, in which a rating of 4 indicated the highest level of anxiety, with a total potential score of 20–80; (2) a 20-item depression questionnaire (DQ) [34], with each question given a weighted score of 1 to 4, with a rating of 4 indicating the highest level of depression. The total possible score ranged from 20 to 80.

The exclusion criteria were as follows: (1) a history of psychiatric or neurological disorders, including a diagnosis of depression; (2) oral contraceptive use within the past year; and (3) intensive regular exercise habits in the past three months. All participants had normal or corrected-to-normal vision and agreed to refrain from alcohol consumption for at least 48 h and caffeine/tobacco use for at least 12 h before the experiment. Participants completed a short visual acuity test before the experiment.

Participants underwent two ESTs: the first measurement was performed before aerobic exercise, and the second was performed during a 90 min resting period after the exercise was terminated. Written informed consent was obtained from all participants prior to the experiment. Participants were informed that they could withdraw from the experiment at any time. All experiments were performed in accordance with the guidelines and regulations of the Institutional Review Board of Chang Gung Memorial Hospital (201004099B0C601) (Figure 1).

### 2.2. Aerobic Exercise

All participants were instructed to perform aerobic exercise on the treadmill for 20 min at a consistent speed of 25.6–28.8 mph (16–18 km/h). Adequate warm-up and cool-down periods and progressive and gradual increments in exercise duration and energy expenditure were implemented according to the recommendations. The warm-up (5 min of a 1% gradient) was followed by graded practice exercises of 5 min each with 3% and 4% grades and a cool-down phase for the last 5 min with grades of 1–2%. An estimate of a person’s maximum age-related heart rate was obtained by subtracting the person’s age from 220 (HR_max_; calculation by 220 minus age). The HR_max_ was recorded and used to check exercise intensity for both groups. Trained nurses ensured the safety of the experimental procedures. 

### 2.3. Pre- and Post-Exercise ESTs

The materials and parameters of the ESTs were designed and modified using a colleague’s previous report in which three different categories of stimuli were created [35]. Each category included 10 different words that were negative, neutral, or positive. The words were displayed in four different colors (yellow, blue, green, and red) in a 1 cm Courier style font against a white background on a computer screen. Each emotion word was presented five times for each color. Each trial proceeded as follows: a fixation point “+” appeared at the center of the computer screen for 0.5 s. The fixation point was then replaced by a stimulus word that remained on-screen for 3 s (unless the response of the participant was faster, in which case the display time was reduced), followed by another fixation point corresponding to an inter-stimulus interval of 2 s and duration of 0.5 s. Participants were asked to classify the color of each word while ignoring its meaning as quickly and accurately as possible by pressing the keyboard using their right index finger. The EST was used to assess attentional bias by examining the differences in RTs and ERs in response to the three emotion word categories (positive, negative, and neutral). The positive, negative, and neutral word lists were matched for word length, frequency, and emotionality (positive and negative words only). The negative word list consisted of general threat-related terms (e.g., “emergency”, “cancer”, or “sneer”).

Our modified EST materials are derived from previously well-validated and reliable sources, and they have also been approved as applicable for the Chinese population [36,37,38]. The protocols were piloted and established before the official experiment to ensure the best possible avoidance of sleepiness, especially with respect to post-exercise fatigue. Evidence reports that emotional Stroop effect (ESE) studies show sustained effects due to habituation resulting from repeated exposure to emotional stimuli [39]. All participants were exposed to 60 trials.

### 2.4. Sample Size and Statistical Analysis

We used G*Power software (version 3.1.9.7) to determine the number of samples. We hoped that the effect size would reach a medium level of 0.13. Cronbach’s alpha was generally 0.05, and the power was set to 0.8. Comparing the differences between the two groups and the three emotion levels, the correlation coefficient between the pre- and post-exercise tasks was approximately 0.9, and the number of output samples was 22.

Descriptive statistics were initially performed to analyze the obtained data, including the calculation of the average age, exercise intensity, and DQ and AQ scores for the two groups. RTs and error rates (ERs) were measured for selecting the color of emotion words (neutral, positive, and negative) both pre- and post-exercise for each group. Two independent sample *t*-tests were then performed to examine differences in DQ and AQ scores between the groups. The independent variables included groups (Peri-MG and YG) and periods (pre- or post-exercise). The dependent variables were the RTs and ERs when selecting the color of words. Mixed-effects models for repeated measures were used to assess differences in RTs and ERs for ESTs both within and between groups. These models enabled the evaluation of differences in baseline scores between the two groups, differences between baseline and post-exercise scores within each group, and the assessment of whether these differences varied between the groups. Bonferroni’s post hoc *t*-tests were performed using ANOVA for multiple comparisons if the main effect was found to be statistically significant (*p* < 0.05). In addition, two independent sample *t*-tests were conducted to determine whether the differences between the groups were statistically significant. A pooled *t*-test was also performed to compare the variance of the RTs and ERs by calculating the differences (pre-exercise minus post-exercise) for emotionally laden words to assess the presence or absence of a differential impact between the two groups. Statistical analyses were conducted using SPSS Version 26.0.

The purpose of this study was to assess the effects of exercise on executive function in perimenopausal women based on the results of the EST, with a specific focus on inter-group comparisons. We did not systematically assess the effects within each category of words (positive, negative, and neutral) in this experiment. Correlation analysis will be conducted on the basis of the DQ and AQ scores if significant inequalities between groups are observed.

## 3. Results

### 3.1. Participant Characteristics

We included 27 healthy participants in this study. The mean ages of the 14 perimenopausal women (Peri-MG) and 13 young women (YG) were 49.23 ± 2.16 and 19.12 ± 0.60 years, respectively. The average exercise intensity was 74% and 60.5% in the Peri-MG and YG.

### 3.2. Neuropsychological Assessment

The DQ scores (mean ± SD) of the Peri-MG and YG were 32.54 ± 12.59 and 34.85 ± 5.57. The AQ scores were 37.82 ± 7.19 and 36.81 ± 4.98 for the two groups. There were no significant differences in the DQ and AQ scores between the two study groups (*p* all > 0.05), which may be attributed to the eligibility screening performed by participants during recruitment.

### 3.3. Differences in RTs and ERs

Mixed model statistics of the RTs (1/10 ms) and ERs of all participants are shown in Table 1a,b. Baseline comparisons between Peri-MG and YG (Table 1b, columns 4 and 5) showed unequal baseline RTs for selecting the color of the three words (*p* all < 0.001). However, these baseline differences were controlled using the mixed model procedure for the differences between the two groups. ERs in the Peri-MG and YG were equal for neutral and positive words at baseline (*p* all >0.05), but a significant increase in ERs with negative words was found in the Peri-MG compared to the YG (*p* < 0.05).

Post-exercise, there were significant decreases in RTs within the Peri-MG for all three word categories (positive, negative, and neutral) (*p* all <0.01) and within the YG for neutral and positive words compared with pre-exercise values (Table 1a, columns 4, 5, 8, and 9).

Comparing the two groups (Table 1b, columns 6 and 7) revealed differences in the RTs and ERs in all three categories (*p* < 0.001 and *p* < 0.05, respectively). Significantly faster RTs were observed post-exercise in the YG (Table 1b; columns 8 and 9). Independent *t*-test comparisons did not reveal any significant differences between the post-exercise ERs of the two groups (*p* > 0.05).

### 3.4. Different Effects of Exercise on EST Performance

Compared with the YG, the Peri-MG showed significant improvement in RTs (that is, with reduced RTs) post-exercise for positive and negative words (*p* all < 0.05), but not for neutral words (*p* > 0.05) (Table 2, Figure 2). There were no significant differences in ERs post-exercise between the two groups for all three categories. Table 2 shows the RT and ER differences in the Peri-MG and YG. There were significantly greater RT improvements in the Peri-MG for positive (t (25) = 2.723; η^2^ = 0.229; *p* = 0.012) and negative emotions (t (25) = 2.335; η^2^ = 0.182; *p* = 0.027), but there was no significant difference in ERs between the two groups (*p* > 0.05).

## 4. Discussion

This study investigated the effect of aerobic exercise on the executive function in perimenopausal women using ESTs. To date, no study has directly compared the impact of exercise on the emotional aspects of Stroop performance in perimenopausal and young women. We found that the Peri-MG exhibited significantly longer RTs than the YG during the baseline and post-exercise tests (Table 1b). The perimenopausal women exhibited a more significant improvement in RTs (i.e., shortened RTs) through exercise modification for positive and negative words but not for neutral words as compared with the YG (Table 2; Figure 2). 

The ability to ignore irrelevant information has been suggested to be a specific executive function [40]. Impaired RTs when sifting through information relevancy are often a sensitive indicator of cognitive change. The Peri-MG exhibited cognitive flexibility and reduced bias to salient cues, which were significantly more sensitive to alterations from exercise. This in turn has implications for understanding the emotional vulnerability of perimenopausal women.

### 4.1. Exercise and Improved RTs during ESTs

Our findings showed that exercise shortened RTs during ESTs for both groups, except in negative words for the YG. The effect of aerobic exercise on executive function was similar in both groups. Numerous studies have shown that exercise can improve cognitive performance and that acute exercise improves participants’ RTs in a Stroop task (ST) [41]. For example, exercise leads to improved performance on the Stroop color–word interference task [42]. Older adults show the same exercise-related improvement as younger adults [43]. Moreover, acute moderate exercise can elicit increased dorsolateral prefrontal activation and improve cognitive performance during an ST [44]. Exercise increases catecholamine concentrations in the brain, resulting in faster processing and improved executive function [45,46]. Furthermore, aerobic exercise can increase blood flow and oxygen supply to the skeletal muscles and brain, which, in turn, affects RTs. The activation of these cognitive–motor connections may be a key factor in providing effective resources for cognitive performance tasks. This paper demonstrated similar significant reductions in ESE post-exercise in both the Peri-MG and YG. Although several studies have proven that exercise improves cognitive flexibility [47], our study specifically focused on a psychologically vulnerable population (i.e., perimenopausal women) and the effectiveness of exercise during this period, as assessed using the EST.

### 4.2. RTs in the Peri-MG

Significantly longer RTs were observed in the Peri-MG during the pre- and post-exercise measurements in response to the three word categories compared with the YG (*p* < 0.001). In this context, the Stroop effect is the mainstay of research on age-related differences in selective attention, automaticity, inhibitory processes, and executive control. The greater ESE observed in older adults can be attributed to age-related deficits in specific cognitive processes [48]. Both sensory perception and processing speed decline with age, thus affecting test performance in several cognitive domains. Deterioration in cognitive abilities such as instant processing speed, executive functions, and episodic memory has been observed in otherwise healthy elderly individuals [49,50]. This deterioration is substantially related to the neuroanatomical differences in the brain characteristics of younger and older participants [49,51,52]. The age-related differences involved in attentional control in the prefrontal cortex suggest a decrease in the ability of older adults to flexibly allocate attention to increasing attentional demands compared with younger adults [53,54]. Furthermore, the cessation of hormonal secretion during menopause in middle-aged women is a cause of mild cognitive impairment or cognitive function disturbance [5]. These perspectives are compatible with our EST results.

### 4.3. Exercise and ER Alteration

ERs increased significantly for negative words in the Peri-MG compared with the YG during the baseline (*p* < 0.05), but this difference was absent in the post-exercise test (Table 1b). This result was not observed for positive and neutral words. The ESE reflects an individual’s sensitivity to the emotional valence entailed in stimuli [55,56]. The perimenopausal period is a time of increased vulnerability to cognitive decline, depressive symptoms, and mood disturbances [57]. Cognitive processes can be modulated differentially according to one’s affective traits. Perimenopausal women have higher rates of anxiety and depressive symptoms than premenopausal women [7]. Individuals with higher characteristics of anxiety or depression, including preclinical clients, were relatively more susceptible to negative emotional stimuli [22], which may explain the significantly higher baseline ERs in the Peri-MG. Exercise plays a crucial role in positively improving neuropsychological functions [14,58]. An individual in a better mood could then better avoid negative word interference [25]. The Peri-MG may have transferred innate mood states through exercise intervention, leading to the absence of differences in ERs between groups in the second test. These explanations may explain the changes in ER variances during the two tests. The Peri-MG may have transferred innate mood states through exercise intervention, leading to the absence of differences in ERs between groups in the second test. These explanations may explain the changes in ER variances during the two tests. However, a more precise refinement and establishment of the psychological effects of exercise are necessary for future research. Aerobic exercise alters cognitive bias for negative words and reduces the ESE, demonstrating the benefits of exercise on executive function in perimenopausal women.

### 4.4. Positive and Negative Word RT Improvement

Compared with the YG, the Peri-MG showed significant improvement in RTs (i.e., a reduction in RTs) with exercise modification for positive and negative words (*p* < 0.01), but not for neutral words (*p* > 0.05) (Table 2). ESTs have been extensively used to investigate attentional processes. Emotionally salient words often preferentially capture attentional resources over neutral information [59,60]. Moderate exercise generally enhances attentional resources related to perceptual processing [61]. In our study, the average exercise intensity was 74% and 60.5% in the Peri-MG and YG. We know that the exercise intensity affects reaction time [62]. Remarkably, the promotion of EST performance through reduced RTs was specific to positive and negative words in the Peri-MG; however, no improvement was observed in neutral words in the YG (Table 2).

Exercise is associated with increased executive function, as well as decreased attentional bias toward emotional and non-emotional words for the Peri-MG, as mentioned above. The pooled *t*-test demonstrated that exercise resulted in a larger reduction in the ESE for emotionally laden words (positive and negative) in the Peri-MG than in the YG (*p* < 0.05; Table 2). In other words, the Peri-MG showed more cognitive flexibility when incorporating exercise, as illustrated by the significant decrease in emotional interference. This observation is likely related to the differences in exercise–cognition interaction, depending on the biological phenotype in question. Exercise enhances the functional capabilities of the brain, which increases brain-derived neurotrophic factor (BDNF) levels via the serotonin–BDNF loop, which is particularly effective in mediating affective neurocircuitry and psychological symptoms [63]. Hormonal variations can influence emotional processing via neuropsychological factors [64], such as the combination of emotional vulnerability and greater susceptibility to exercise modulation in the Peri-MG. The ESE can be interpreted as an involuntary inability to effectively inhibit automatic reading processes [25]. This accounts for the self-resilient and adaptive transition reinstatement for regulatory capabilities but natural psychological vulnerability of the Peri-MG.

Meanwhile, the Peri-MG exhibited more efficient executive control processes in response to emotional words and disengagement from salient cues from exercise. It is worth noting that there is often an attentional bias toward emotion-related words in individuals with anxiety, depression, or other psychological disorders [65]. In this context, the EST documents emotion-related regulation mechanisms [25]. Acute exercise is one of the most effective behavioral techniques used for mood self-regulation in healthy populations [12]. The ability to focus on critical environmental elements while ignoring irrelevant salient information is essential for adaptive behavior and psychological well-being. Moreover, the difference in the ERs returned to post-exercise levels, as mentioned earlier. In summary, our study suggests that exercise can effectively reduce the bias toward salient cues in perimenopausal women, improving executive function and potentially alleviating mood disturbances.

### 4.5. Strengths

To date, no study has directly compared the impact of exercise on the emotional aspects of Stroop performance in perimenopausal and young women. Diverging from previous evidence, we have presented the potential benefits of acute exercise in psychologically vulnerable groups, such as heightened sensitivity in processing salient stimuli, as observed in the EST. We approach multidisciplinary domain interpreting cognitive frailty relates in research results, serving as a foundation for future development efforts in preventative medicine research and healthcare, thereby contributing to advancements in women’s health [4].

### 4.6. Limitations

Our study has several limitations. First, the sample size was small, though our within-subject comparisons might nevertheless have provided greater statistical power for this study. Second, it is important to note that this was only a pilot experiment, hence the limited number of subjects. Thirdly, this study observed the acute effects of exercise. Moving forward, a greater sample size and more rigorous or varied exercise interventions, such as chronic training, are needed to achieve more effective and robust experimental statistical effects.

## 5. Conclusions

Our findings indicate that aerobic exercise can enhance executive control performance in perimenopausal women. Specifically, aerobic exercise significantly reduced emotional Stroop interference (ESE) for emotionally laden words, underscoring its potential to mitigate cognitive decline and reduce the risk of vulnerability to psychological disorders in this population.

## Figures and Tables

**Figure 1 behavsci-14-00338-f001:**
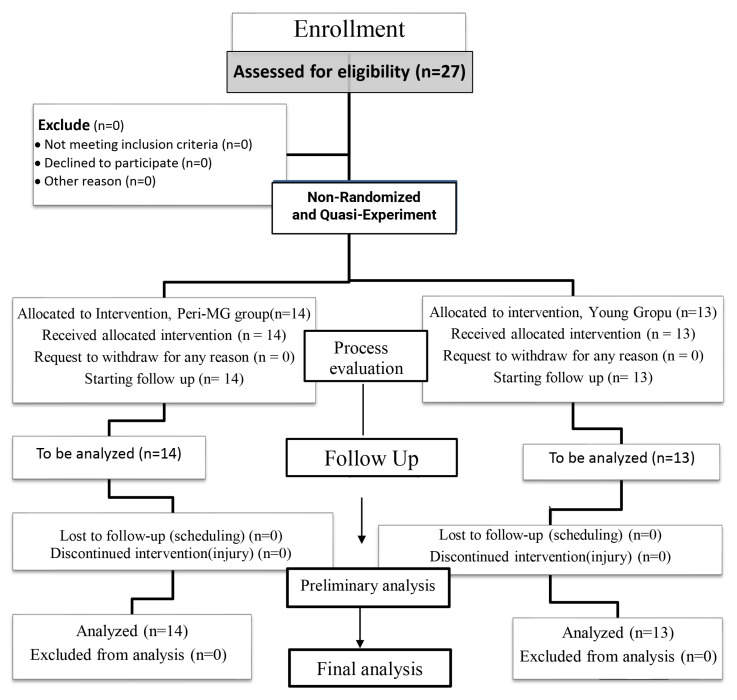
Flowchart of the study design and participants (based on CONSORT).

**Figure 2 behavsci-14-00338-f002:**
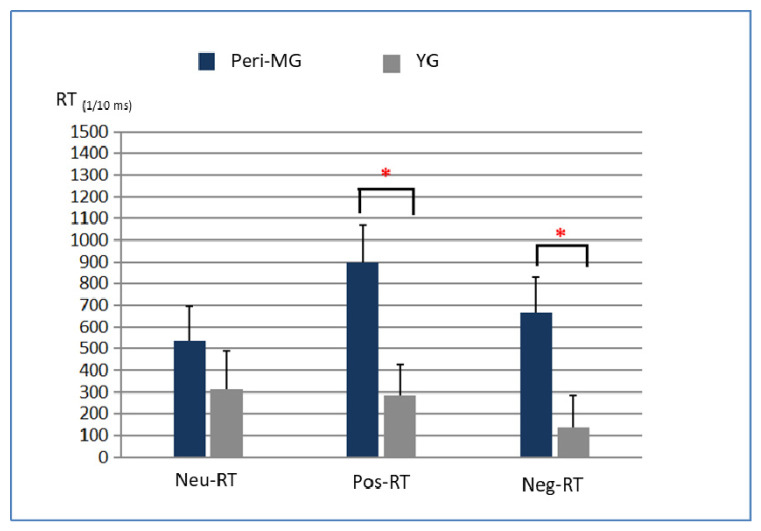
Comparison of post-exercise EST performance of the two groups. ** = significant at the level of 0.05*; rectangular and error bars indicate means and standard errors across participants.

**Table 1 behavsci-14-00338-t001:** (**a**) Mixed models statistics for within-group, baseline, and between-group differences in RTs and ERs for emotion words for Peri-MG and YG. (**b**) Within-group, baseline and between-group differences for two group.

(a)								
	Perimenopause Group (Peri-MG) Mean Differences (MDs)				Young Group (YG) Mean Differences (MDs)			
	Baseline Mean (SD)	Post-Exercise Mean (SD)	MDs	t (df = 13)	Baseline Mean (SD)	Post-Exercise Mean (SD)	MDs	t (df = 12)
**Neutral-RTs**	8337 (980)	7799 (929)	537	3.364 **	6604 (752)	6292 (526)	312	1.787 *
**Positive-RTs**	8586 (1108)	7687 (1000)	899	5.229 ***	6672 (760)	6388 (613)	284	1.976 *
**Negative-RTs**	8377 (1119)	7715 (1011)	662	4.021 ***	6580 (664)	6444 (614)	136	0.914
**Neutral-ERs**	0.143 (0.363)	0.071 (0.267)	0.071	0.563	0.077 (0.277)	0.154 (0.376)	−0.077	−0.562
**Positive-ERs**	0.357 (0.277)	0.214 (0.426)	0.143	0.693	0.231 (0.439)	0.077 (0.277)	0.154	1.000
**Negative-ERs**	0.571 (0.938)	0.143 (0.363)	0.429	1.578	0.077 (0.277)	0.077 (0.277)	0.000	0.000
**(b)**								
	**Within-Group** **Mean Differences**		**Baseline** **Mean Differences**		**Between-Group** **Mean Differences**		**Second (Post-Exercise)** **Mean Differences**	
	**MDs**	**F (df = 1.25)**	**MDs**	**t (df = 25)**	**MDs**	**F (df = 1.25)**	**MDs**	**t (df = 25)**
**Neutral-RTs**	424.691	12.930 ***	1733	5.126 ***	1620.47	2440.099 ***	−1507.80	−5.13 ***
**Positive-RTs**	591.683	27.417 ***	1914	5.193 ***	1606.440	2008.113 ***	−1298.70	−4.03 ***
**Negative-RTs**	399.233	12.781 ***	1797	5.022 ***	1534.252	2029.724 ***	−1271.31	−3.91 ***
**Neutral-ERs**	−0.003	0.001	0.066	0.527	−0.008	7.155 *	0.082	0.515
**Positive-ERs**	0.148	1.301	0.126	0.598	0.132	12.771 **	−0.137	0.328
**Negative-ERs**	0.214	2.006	0.494	1.827 *	0.280	8.729 **	−0.066	0.603

* = significant at the level of 0.05, ** = significant at the level of 0.01, *** = significant at the level of 0.001. RTs: reaction times (1/10 ms); ERs: error rates; MDs: mean differences; SD: standard deviation.

**Table 2 behavsci-14-00338-t002:** Pooled *t*-test for the promotion of EST performance of the two groups post-exercise.

Catagory	Group	Mean (SD)	t (df = 25)	Sig.	Eta Squared
Neutral-RTs	Perimenopause	537 (598)	0.954	0.349	0.035
	Young	312 (630)			
Postive-RTs	Perimenopause	899 (644)	2.723	** *0.012* ** ***	0.229
	Young	284 (518)			
Negative-RTs	Menopause	662 (616)	2.355	** *0.027* ** ***	0.182
	Young	136 (537)			
Neutral-ERs	Perimenopause	0.071 (0.47)	0.796	0.434	0.025
	Young	−0.076 (0.494)			
Postive-ERs	Perimenopause	0.143 (0.770)	−0.042	0.967	0.000
	Young	0.153 (0.555)			
Negative-ERs	Perimenopause	0.429 (1.02)	1.416	0.169	0.074
	Young	0 (0.408)			

** = significant at the level of 0.05*; RTs: reaction times =1/10 ms; ERs: error rates; MDs: mean differences (calculated via pre- minus post-exercise); SD: standard deviation.

## Data Availability

The datasets generated during and/or analyzed during this current study are available upon reasonable request.

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
