# Peer review of "Effect of Physical Exercise on Executive Functions Using the Emotional Stroop Task in Perimenopausal Women: A Pilot Study"

_behavsci, 2024, doi:10.3390/bs14040338_

Round 1
Reviewer 1 Report
Comments and Suggestions for Authors
I would like to thank the authors for their manuscript.
Below, some minor points need to be considered.
1. In the abstract section, Please change the phrase (P all < 0.05) to P<0.05. It is not statistically correct to put it that way.
2. Please provide a CONSORT diagram to the section 2.1. Study Design and Participants.
3. Are there any studies regarding the validity and reliability of the EST tool? Please provide at least two references in the section 1.3. Emotional Stroop tasks.
Author Response
Dear Reviewer,
Thank you for reading and reviewing our manuscript. The critical and constructive comments have certainly helped us improve our content. We have revised our manuscript accordingly, and all revisions have been highlighted within the document using red-colored text. Please receive the attachment.
Sincerely,
Ren-Jen Hwang

Reviewer 2 Report
Comments and Suggestions for Authors
General Evaluation
This study investigates the influence of exercise on cognitive processes among young and perimenopausal women. The title underscores the focus on cognitive function and age-specific considerations. The abstract succinctly outlines the study's purpose, methodology (including participant selection, settings, measurements, and analytical approaches), key findings, and principal conclusions.
Introduction
The introduction of this study should provide a robust foundation by elucidating essential conceptual terms to understand the focus of the research on the impact of exercise on cognitive processes in young and perimenopausal women. Central to this understanding are the definition of concepts such as executive functions, neurocognitive function, cognitive control, cognitive processing. These concepts are used indiscriminately throughout the article.
By clarifying these conceptual terms in the introduction, the study aims to establish a clear theoretical framework and provide a solid foundation for understanding the subsequent investigation into the effects of exercise on cognitive processes in women at different stages of life. This reinforcement enhances the clarity and coherence of the research objectives, facilitating a more comprehensive understanding of the significance and implications of the study.
Methods
The methods, tools, and procedures warrant some consideration to address the research question effectively.
While comparing young women with perimenopausal women may be relevant to the study, a sample of this size (27 women) may not be sufficient to capture the full complexity and variability associated with different ages and life stages. With a small sample, it may be more challenging to effectively control other variables that could influence the results, such as fitness level, medical history, lifestyle, and other potentially relevant factors.
Another point relates to the emotional Stroop tasks. The Emotional Stroop Task is particularly valuable if the aim is to investigate how exercise affects emotional processing and emotional regulation. However, the Stroop Test may be more appropriate if the authors are interested in investigating cognitive abilities, regardless of emotional content. Applying the Emotional Stroop Task introduces another issue to control, which is the effects of emotion on cognitive control. This is particularly relevant given the small sample size combined with two distinct sample groups, differing in age and emotional symptoms.
Results
The results are presented in tables and figures displaying the data appropriately.
Discussion
The results are discussed and compared with other investigations. However, it is essential to interpret the findings cautiously, considering the small sample size and the potential influence of age differences on the results.
Conclusions
The results address the objectives.
Author Response

(The authors gave the same response as above.)
